# The Yield Curve as a Leading Indicator: Accuracy and Timing of a Parsimonious Forecasting Model

**Knut Lehre Seip** [1,*] **and Dan Zhang** [2]

1   Art and Design, Faculty of Technology, Oslo Metropolitan University, 0167 Oslo, Norway
2   Oslo Business School, Oslo Metropolitan University, 0167 Oslo, Norway; danzhang@oslomet.no
*   Correspondence: knut.lehre.seip@oslomet.no; Tel.: +47-67238816

**Abstract:** Previous studies have shown that the treasury yield curve, T, forecasts upcoming recessions when it obtains a negative value. In this paper, we try to improve the yield curve model while keeping its parsimony. First, we show that adding the federal funds rate, FF, to the model, GDP = f(T, FF), gives seven months vs. five months warning time, and it gives a higher prediction skill for the recessions in the out-of-sample test set. Second, we find that including the quadratic term of the yield curve and the federal funds rate improves the prediction of the 1990 recession, but not the other recessions in the period 1977 to 2019. Third, the T caused a pronounced false peak in GDP for the test set. Restricting the learning set to periods where T and FF were leading the GDP in the learning set did not improve the forecast. In general, recessions are predicted better than the general movement in the economy. A "horse race" between GDP = f(T, FF) and the Michigan consumer sentiment index suggests that the first beats the latter by being a leading index for the observed GDP for more months (50% vs. 6%) during the first test year.

**Keywords:** term structure; federal funds interest rate; GDP; forecasting; economic growth; aggregate productivity

## 1. Introduction

An accurate forecast of economic growth and upcoming economic recessions is crucial to households, businesses, investors and policymakers. There is a vast literature on forecasting economic growth and recessions based on macroeconomic indicators, among which the treasury yield curve has often been cited as a leading indicator, with inversion of the curve being a signal of a recession. For example, Estrella and Hardouvelis [1], Estrella and Mishkin [2] and Estrella and Mishkin [3] explained how the yield curve significantly outperforms other financial and macroeconomic indicators in predicting recessions two to six quarters ahead.

Studies also provide empirical evidence showing that the domestic government bond yield spreads are the best recession predictor for output growth and recessions (see, for example, Duarte, Venetis [4] and Nyberg [5]). In addition, recent studies on predicting recessions find that the yield curve consistently outperforms even professional forecasters who have a rich set of indicators and forecasting tools available to them (Rudebusch and Williams [1] and Croushore and Marsten [6]).

Our analysis differs from earlier studies of forecasting economic growth and recessions by focusing on both parsimony and the timing and accuracy of the predictions. Recent studies of forecasting economic growth and recessions often use complex mathematical models and a large set of financial and macroeconomic variables. We examine if the predictions based on the treasury yield curve can parsimoniously be improved to give a longer warning time and better accuracy. The added value of our study is fivefold. *First*, we examine the benefit of including the federal funds rate (FF) and its interactions with the treasury yield curve (T) in a multiple equation regression. *Second*, we restrict the learning

set to portions of the time series where T and FF lead the GDP. *Third*, we compare six possible models to see which models gives the best prediction of the general movements and of the five NBER recessions during our study period 1971 to 2019. Fourth, we show visually how time series predicted with T alone would differ from predictions with FF alone. This allows us to identify events that are caused by a particular variable. *Finally*, we arrange a "horse race" between our prediction GDP = f(T, FF) and the Michigan consumer sentiment index (MCSI), a frequently used index for forecasting movements in the GDP one year ahead.

Predicting business cycles during their general development, and during recessions and recoveries, is of clear interest to policymakers and market participants. Policymakers may respond to the forecasts by adjusting their policies for economic expansions and contractions. They could also use the forecasting model to quantify the economic impact under different scenarios. Market participants may utilize the forecast to assess risks and adjust their investment strategies accordingly. Our forecasting model that combines parsimony, predictive accuracy, timing and ease of estimation could benefit policymakers and market participants.

In the rest of the manuscript, we present the literature review and develop the hypothesis in Section 2. In Section 3, we describe our data and present the methodology and our testing procedures. The results are shown in Section 4, and we discuss the results in Section 5. Finally, we discuss policy implications and conclude the findings in Section 6.

## 2. Literature Review and Hypotheses

A large body of literature studies the predictive power of financial and macroeconomic leading indicators for real output growth and recessions. Most prominently, Estrella and Hardouvelis [2], Estrella and Mishkin [3] and Estrella and Mishkin [4] documented that the slope of the treasury yields curve has strong predictive power for US output growth and US recessions at horizons of up to eight quarters. Chauvet and Potter [7] examined further extensions of the yield curve probit model, including a business cycle-dependent model, a model with autocorrelated errors and combinations of these extensions. They found evidence in favor of the more sophisticated models that allows for autocorrelation and multiple breakpoints across business cycles.

In addition to the empirical evidence in the US, other works have documented the strong predictive power of the government bond yield spreads for output growth and recessions internationally. For example, Duarte, Venetis and Payà [5] confirmed the ability of the yield curve as a leading indicator to predict recessions in the European Monetary Union. Nyberg [6] examined recessions in the USA and Germany and showed that the domestic term spread remains the best recession predictor.

While the yield spread has long been recognized as a good predictor of recessions, it seems to have been largely overlooked by professional forecasters. Rudebusch and Williams [1] found that the yield curve consistently outperforms professional forecasters in predicting recessions. This is puzzling given that professional forecasters have a rich set of indicators and forecasting tools available to them. Lahiri, Monokroussos [8] and Croushore and Marsten [7] confirmed that Rudebusch and Williams's [1] findings are robust, including augmenting the model with more macroeconomic factors, the use of different sample periods, the use of rolling regression windows and various alternative measures of real output. Yang [9] found the yield curve to have a well-performing ability to forecast the real GDP growth in the USA, compared to professional forecasters and time series models.

In our study, we try to make the best use of the predictive power of the yield curve as documented in the literature and ask whether we could improve the yield curve model while keeping the model as parsimonious as possible. We developed four hypotheses that we will test empirically. Our first hypothesis (H1, the baseline model) is that the treasury yield curve alone will explain most of the variation in the GDP. All the papers

discussed above highlight the singular importance of the treasury yield curve as a predictor of recessions and justify our use of this indicator as the benchmark predictor variable.

Economic growth, recessions and interest rates are all endogenous and any association among them could be considered a reduced form correlation. Estrella and Hardouvelis [2] tested a model with both the yield curve and the federal funds rate included. Their results showed that a higher real federal funds rate today is associated with a lower growth in the future real output. Bauer and Rudebusch [8] documented that accounting for dynamic changes in the equilibrium short rate is essential for forecasting the yield. Galbraith and Tkacz [9] found that the yield curve–output relation might not be linear and its predictive content might have asymmetric effects. The work by Venetis, Paya and Peel [10] showed that the relationship is stronger when past values of the yield spread do not exceed a positive threshold value. Therefore, our second hypothesis, H2, is that predictions of the GDP will improve if we augment the yield curve model with the federal funds rate. Specifically, we will include second-order interactions as additional variables because the T and the FF may give complementary information on the state of the economy when recessions are not imminent. Furthermore, the relationship between the prediction algorithm and GDP may not be stable over time and it may be subjected to nonlinearities (see, for example, Galbraith and Tkacz [10] and Venetis, Paya [11]).

Concerns have been raised that the predictive performance of the yield curve model may be time-variant, and that predictive regressions based on the yield spread may suffer from parameter instability, e.g., Estrella, Rodrigues [12] and Giacomini and Rossi [13]. Estrella, Rodrigues [12] studied the United States (and Germany) and showed that there is some evidence of instability in the real growth models for the United States, whereas the recession models are generally stable. Giacomini and Rossi [11] documented the existence of a forecast breakdown, whereas during the early part of the Greenspan era, the yield curve emerged as a more reliable model to predict future economic activity. Therefore, our third hypothesis, H3, is that the explained variances between the predicted and the observed GDP will be higher if the forecasting models are restricted to the time window where the yield curve and the federal funds rate are leading variables to GDP. The rationale is that the leading relations between the yield curve, the federal funds rate and GDP may change with time, e.g., Schrimpf and Wang [14], and in some periods, the information carried by the yield curve and the federal funds rate is not available before the market makes its decisions. Therefore, it may distort the estimation of the parameters in the prediction equations if we include observations of the independent variables that occur after the GDP has changed.

Our fourth hypothesis, H4, is that recessions would be better predicted than the overall GDP. The rationale is that Seip, Yilmaz [15] found for the German economy from 1991 to 2016 that recessions had a higher probability to be predicted correctly than the overall GDP time series by two German sentiment indexes. In general, there may be periods in the development of the GDP that are better predicted than others, and algorithms that are robust across macroeconomic breakpoints should be preferable.

### 3. Data and Methods

*3.1. Data*

We used the real GDP as a proxy for real economic growth and identified recession periods using National Bureau of Economic Research (NBER) definitions. The recessions in the USA during the period 1977 to 2019 are shown in Table 1.

We measured the yield curve, T, as the difference between the 10-year and 2-year treasury bond yields. This is in the maximum maturity spread range. Long ranges were found to be the best measure of the spread slope by Ang, Plazzesi [16]. We obtained the data for GDP and the difference between the 10-year constant maturity and the 2-year treasury constant maturity from the Federal Reserve Bank of St. Louis (https://fred.stlouisfed.org/series/T10Y2Y, accessed on 20 March 2021). The federal funds rate, FF, was also retrieved from the Federal.

| Recession | Key Factors | Dates | Neg. T Leads Recession, Months |
|---|---|---|---|
| The 1980 recession | The Volcker inflation targeting | January 1980–July 1980 | 19 |
| The 1981–1982 recession | The 1979 energy crisis | July 1981–November 1982 | - |
| Early 1990s recession | 1990 oil price shock | July 1990–March 1991 | 14 |
| Early 2000s recession | The dotcom bubble, the 9/11 attacks and accounting scandals at major US corporations | March 2001–November 2001 | 16 |
| The 2008 recession | The subprime mortgage crisis in US and global financial crisis | December 2007–June 2009 | 21 |
| Average | | | 17.5 ± 3.1 |

Note: data from NBER, National Bureau of Economic Research.

Reserve Bank of St. Louis (https://fred.stlouisfed.org/series/FEDFUNDS, accessed on 20 March 2021). We used economic data from 1977 to 2019M5 at a monthly frequency, that is, 512 entries. As the GDP data are quarterly, we interpolated the GDP data to monthly data. We were able to add data from 2019M6 to 2020M4 for GDP and from 2019M6 to 2020M8 for T and FF. However, as the 2020 recession is ongoing, we only included the recent data points for comparison purposes.

The data were linearly detrended to avoid long-term effects and thereafter centered and normalized to unit standard deviation (There is a concern that detrending might affect forecasting results. However, there is no canonical way to detrend because the effects generated by dynamic chaos may occur in economic as well as ecological time series (Sugihara, et al. [17]. Furthermore, the series may be superimposed of series that are related to different economic processes. We apply linear detrending as it is the simplest form of detrending available. Another alternative, such as taking the first derivative of the series, is not ideal because it introduces much more noise and it shifts the series backward so that, for example, peaks occur before the peak of the raw series). This could be conducted without loss of information because the two series are measured in different units. Last, the data were LOESS smoothed with parameters (f) = 0.1 and (*p*) = 2 to avoid high frequency variability. Here, f is the fraction of the time series that is used as a moving window and *p* is the order of the polynomial function used for interpolation. Figure 1 shows the time series for T, FF and GDP from 1977 to 2019 linearly detrended and normalized to unit standard deviation. The green and bold portions of the T and FF curves will be discussed below.

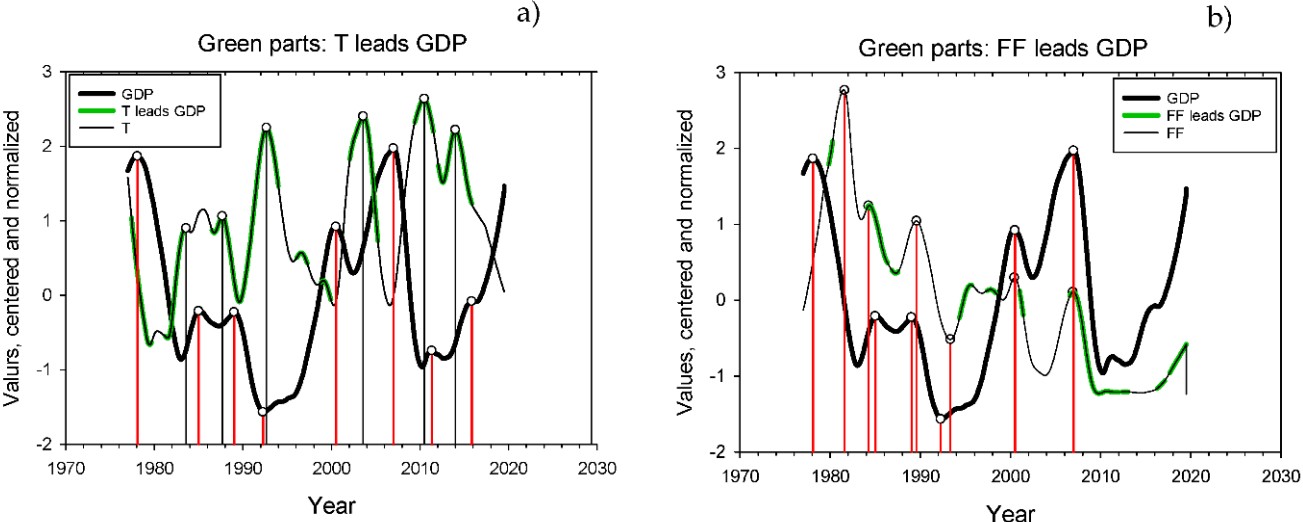

**Figure 1.** The Federal funds interest rate, FF, the Yield curve, T, and GDP. (**a**) Green portions of the curve, T, shows where T leads GDP. (**b**) Green portions of the curve, FF, shows where FF leads GDP. The drop-down lines allow a comparison of peaks and troughs in the two time series. Peaks and troughs are identified from the numerical time series.

### 3.2. Methodology

To identify time windows where the regressors lead the target variable GDP, we applied a lead–lag method to the time series [15,18]. The green curves in Figure 1a,b show where T and FF lead GDP. The drop-down lines show peaks and troughs in the curves. When we restrict the learning set to time windows where either T or FF leads GDP, we use the results depicted in these graphs.

### 3.2.1. Multiple Regression

We applied multiple regressions to the independent variables T, $T^2$, FF, $FF^2$ and T × FF (Although other methods, such as the probit models, are more commonly used Johansson and Meldrum [19] Bauer and Mertens [20], these models focus on recession forecasts, whereas we also compare predicted and observed full series. A recent set of studies use the machine learning framework. Gogas et al. [21], Medeiros et al. [22]. Last, we follow Bauer and Mertens [23]. "Information in the Yield Curve about Future Recessions" Retrieved 17 September 2020, in their adage "correlation is not causation" and examine possible causal links in addition to statistical associations). We conducted regressions with the regressors both shifted and not shifted relative to the dependent GDP.

$$\text{GDP}_t = \alpha_0 + \alpha_1 T_{t+n} + \alpha_2 T^2_{\ t+n} + \alpha_3 FF_{t+n} + \alpha_4 FF^2_{\ t+n} + \alpha_5 T_{t+n} \times FF_{t+n} \tag{1}$$

where $\alpha_i$ is the coefficients to be estimated and n = 0 in the zero shifted alternative. We also tried values of n from −1 to −4. We used the multiple regression algorithm as implemented in SigmaPlot©.

The predictions with only one variable gave a narrow range of values ($\approx$−1, +1; not shown), but the interesting feature is the differences between the observed and the predicted time series. We therefore normalized the series to unit standard deviation to correspond in range with the normalized GDP series before calculating the RMSE. The time lags (months) were calculated as the difference between the predicted and the observed turning points for GDP after the observed GDP time series have been slightly LOESS smoothed (f = 0.1, *p* = 2) to avoid sharp peaks in the time series.

We divided our data into a learning set, 1977M1-2004M12, and a test set, 2005M1 to 2019M5. The coefficients of the equations were determined by applying the equations to the learning set. The forecasting skill was determined by calculating the RMSE between the calculated and observed time series. Last, we applied the forecasting equations to periods before the five recessions.

### 3.2.2. Comparing GDP to Forecasts

We used two methods to compare GDP to forecasts: the root mean square error (RMSE), and the adjusted explained variance, $R^2$, of the least square regression. The RMSE would give a measure of the difference between the observed and the predicted GDP over a certain period. The $R^2$ would measure the skill of predicting co-movements between the observed and the predicted GDPs. We also report R to maintain the sign of the regression, and we add a sign for the RMSE to see if the predicted GDP is above or below the observed GDP. First, we evaluated the result by calculating RMSE between the predicted GDP ($\text{GDP}_p$) and the observed GDP ($\text{GDP}_o$):

$$\text{RMSE} = \sqrt{(1/T \sum (\text{GDP}_p\text{-GDP}_o)^2)} \tag{2}$$

We conducted a tentative test statistic by regressing GDP to a uniform random distribution using the RAND () function in Excel and calculating the RMSE. Using an autoregressive AR (1) model made the RMSE statistics, on average, larger and worse. We did this for the full series 1977M1 to 2019M5, RMSE (GDP512, R512), the test set 2005M1 to 2019M5, RMSE (GDP175, R175), and for 20 months before the five recessions, RMSE (GDP20, R20). We did this 10 times and calculated the average and standard deviation of the RMSE. Last, we

calculated the RMSE for two stochastic series of the same length as our full time series, RMSE (R512, R512).

The forecasting horizons in our study are 197 to 336 months for the learning set, 175 months for the test set and 20 months for the periods before the recessions.

Using RMSE as a measure of the forecasting skill may be misleading because a series that shows a trough before the observed trough will give a larger RMSE than the one that predicts a trough at exactly the observed time. The latter case is not better from the policymaker's point of view because it would not give an early warning. Second, using the explained variance, $R^2$, or R, of the regression has a similar drawback and also other caveats [24]. To facilitate interpretations, we therefore depict the observed and the predicted GDP time series together.

Robustness is an important part of a leading index and its forecasting skill. We therefore calculated an expression for robustness as the product of prediction skill and the timing of the prediction for all recessions during a certain period. We normalized the robustness measure by normalizing skill and timing across recessions to unit standard deviation. However, the timing measure should be chosen to correspond with the desired lead time for the predictions, e.g., in months or quarters.

## 4. Results

As a piece of backdrop information, we found that the time between the first month that the treasure yield curve became negative and the first month of the last five recessions, 1977–2019, defined by the National Bureau of Economic Research (NBER), was $17.5 \pm 3.1$ months. We then show the results for forecasting GDP with FF and T and their quadratic and multiplicative terms as independent variables. We first examine the full series and the test set. Then, we examine time windows before the recession periods.

### 4.1. GDP as a Function of T and FF

We examine the forecast of GDP made both with the full series for T and FF and with the series restricted to the time windows where T and -FF are leading variables to GDP.

*Unrestricted data for the learning set*. The function GDP = f(T) is shown in Figure 2a. The short, bold line at the bottom right of the figure shows the portion of the time series that was used as a test series. The months 2019M6 to 2020M8 are not included in the test set. The forecasted time series show a fair correspondence with the observed GDP. In particular, the decline in GDP during the recessions in the 1980s, the 1990s and 2008 is reproduced well. However, all recessions, except for the last, belong to the learning set.

The calculated GDP = f(T) was a leading variable to the observed GDP in the test period from 2005M1 to 2009M5 (17 months), and then it became a lagging variable until 2016 and then a leading variable again (see Supplementary Materials 1). Similar dates applied for the other forecasting functions.

Figure 2b shows the results with GDP = f(T, FF). The overall correspondence between the calculated and observed values in the learning set and the test set is not very different from the results for GDP = f(T), but there is a pronounced peak in the test set in 2013M12 that is not reflected in the observed GDP.

We separately calculated the RMSE between the observed and calculated GDP series for the learning set, RMSE-L, and for the test set, RMSE-T. Statistical characteristics are shown in Table 2. The forecasting equations GDP = f(T) and GDP = f(T, FF) in Table 2 rows a and b have an RMSE for the test set of 30% to 57% of the RMSE for GDP paired to a random set, (0.356/1.18 and 0.674/1.18, respectively). However, the regression equation GDP = f(T, FF) explains only a small part, $R^2 = 0.122$, of the association between the GDPo and GDPp.

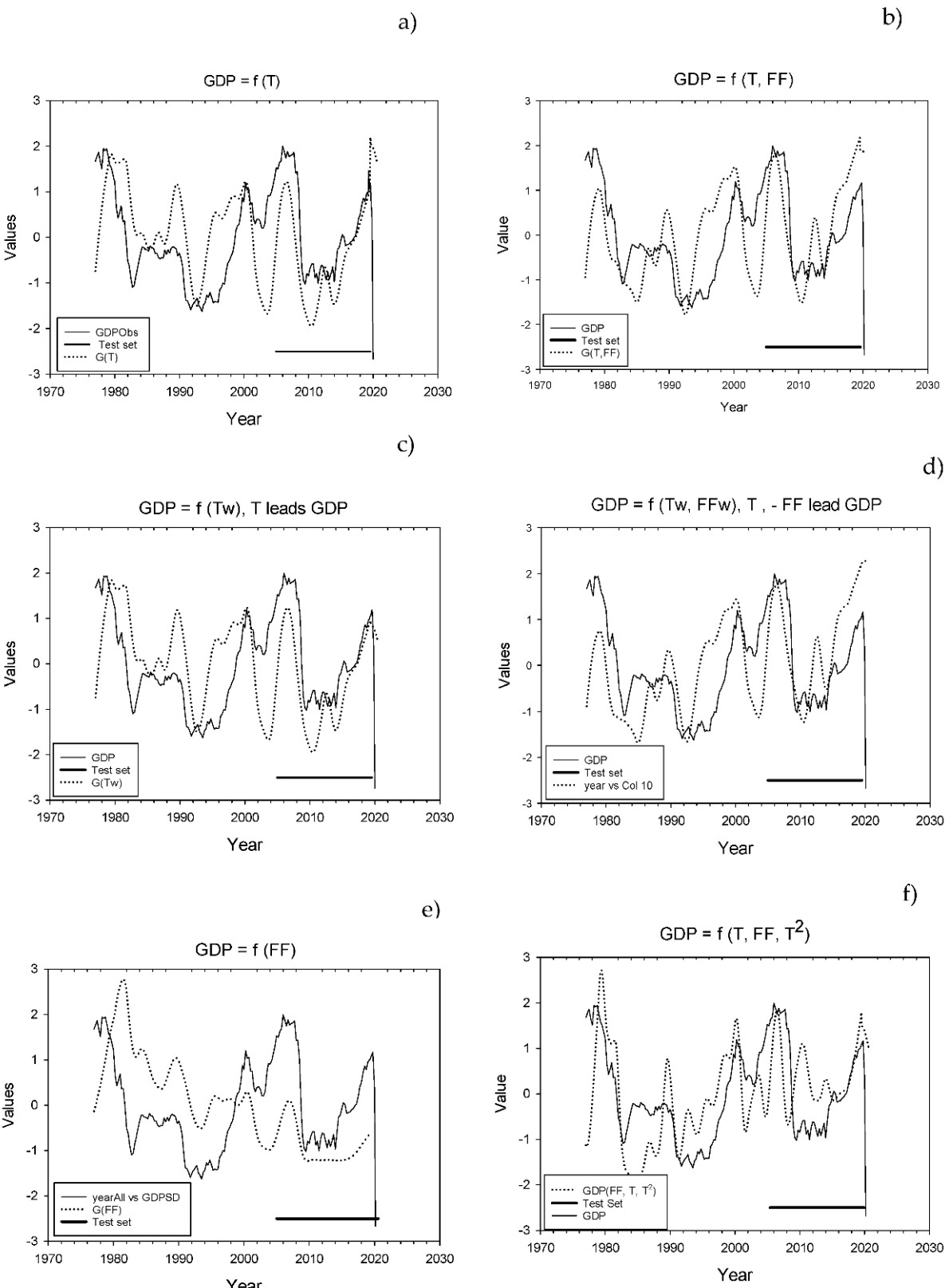

**Figure 2.** Comparison of Observed GDP with calculated and normalized GDP under different restrictions on the variables in the learning set. (**a**) GDP = f(T). (**b**) GDP = f(T, FF). (**c**) GDP = f(Tw), Tw indicate that both time series are restricted to periods when T is leading GDP. (**d**) GDP= f(Tw, FFw), Tw and FFw indicate that GDP, T and FF are restricted to time windows where both T and −FF lead GDP. The thick line to the lower right indicates the test set. (**e**) GDP = f(FF). (**f**) GDP = f(T, FF, $T^2$).

**Table 2.** Regressions for GDPo as a function of treasury yield, T, and federal funds rate, FF.

| | Equation | Learning Set | | Learning Set GDP $_{pred}$ vs. GDP $_{obs}$ | | | | Learning Set | Test Set | Time Lag Month |
|---|---|---|---|---|---|---|---|---|---|---|
| | | n | Set Characteristics | Adj.$R^2$ | $p_1$ | $p_2$ | $p_3$ | RMSE-L | RMSE-T | |
| a | GDP = −0.174 − 0.308 × T | 336 | | 0.082 | <0.001 | | | 1.413 | 0.674 | 5 |
| b | GDP = −0.0796 − 0.320 × FF − 0.538 × T | 336 | | 0.122 | <0.001 | <0.001 | | 0.851 | 0.356 | 7 |
| c | GDP = −0.143 − 0.302 × Tw | 214 | T → GDP | 0.083 | <0.001 | | | 1.424 | 0.663 | 5 |
| d | GDP = 0.113 − 0.829 × FFw − 0.915 × Tw | 197 | T, −FF → GDP | 0.243 | <0.001 | <0.001 | | 0.938 | 0.621 | 5 |
| e | GDP = −0.160 + 0.115×FF | 336 | | 0.008 | 0.057 | | | 1.908 | 1.547 | 1 |
| f | GDP = −0.392 − 0.423 × FF − 0.548 × T + 0.381 × $T^2$ | 336 | | 0.255 | <0.001 | <0.001 | <0.001 | 1.062 | 1.060 | 7 |

Note: There are six equations. $p_1$, $p_2$ and $p_3$ designate the probabilities for the three first independent variables. RMSE-L and RMSE-T are the root mean square error for the learning set and the test set, respectively. The RMSE values were calculated after normalization of the observed and the predicted values to unit standard deviation. The RMSE for the observed GDP vs. the random distribution Rand, 1977 to 2019M5, is 1.334 ± 0.038, and for the test set 2005M12 to 2019, it is 1.179 ± 0.028. For comparison, two uniform random distributions Rand1 vs. Rand2, n = 512, would give RMSE = 1.966 ± 0.138. The arrows in the set characteristic column show that T and -FF lead GDP.

As all three time series are normalized to unit standard deviation, we can use the coefficients in front of the independent variables to express the contribution of T and FF to the prediction of GDP. The T in GDP = f(T, FF) contributes 63% (=0.538/(0.320 + 0.538)) of the explanation to the predicted GDP.

*Restricted data in the learning set.* We restrict the learning set to time windows (w) where T and -FF were leading GDP. The rationale is that the investors that partly determine and partly react to GDP changes may have important prior information. The numerical results are shown in Table 2 rows c and d. The equation in row d, GDP = f(FFw, Tw), has a greater explained variance with respect to the learning set, 0.243, than GDP = f(FF, T), but it only forecasts similarly to Equations (a) and (c), that is, it gives an RMSE value around 60% of the test value (0.621/1.18). Except for model GDP = f(FF) in Equation (e), the forecast equations predicted a recession 5 to 7 months before it occurred.

We used all independent variables, T, $T^2$, FF, $FF^2$ and T × FF, as regressors. Only the variables T, $T^2$ and FF showed significant contributions after forward and backward regression. The result is shown in Figure 2f and in Equation (f) in Table 2.

GDP = f(T, F) shows the overall best forecasts for all five recessions, but the GDP = f(T, $T^2$, FF) equation has a similar prediction skill and it predicts the 1990 recession better. GDP = f(T) and GDP = f(Tw) make the worst predictions.

*Forecasting horizons.* For the prediction curve, we performed calculations at forecasting horizons 1 through 8 quarters, 12, 16, 20, 30, 40, 50 and 60 quarters, Figure 3. The regression coefficient, R, (not adjusted for degrees of freedom) varies between 0.62 and 1.0; however, it ends at 60 quarters (before the 2020 COVID-19 recession), at R = 0.78.

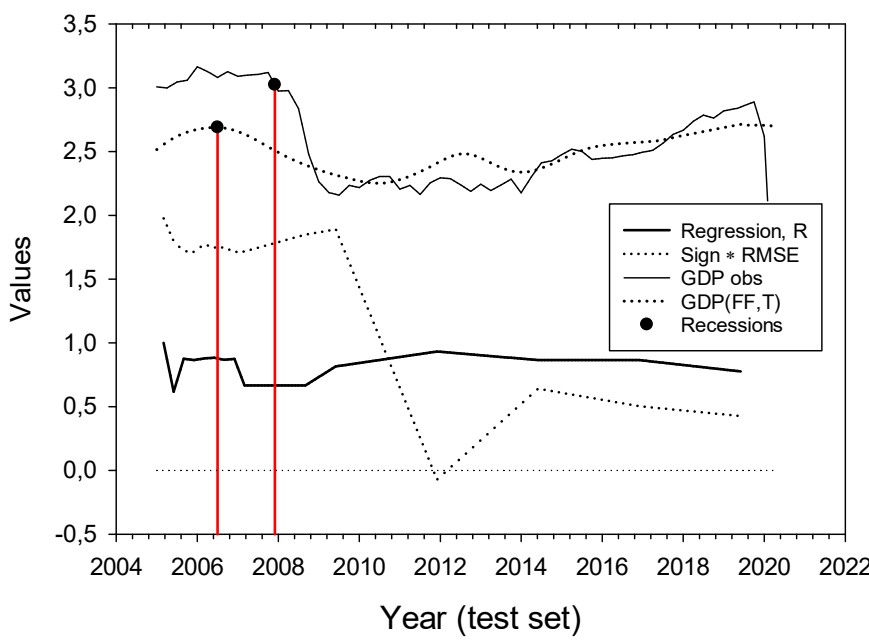

**Figure 3.** Comparison of observed GDP with predicted GDP(T, FF). The drop-down lines show the NBER 2008 recession date (right) and the date when the predicted GDP starts to decline (left). The regression coefficient, R, between observed and predicted GDP is always positive and ends in quarter 60 at R = 0.776, ($R^2$ = 0.60). The average root mean square error is calculated with a sign so that if predictions are below observed values, the RMSE is positive. Regressions and RMSE are calculated at quarters 1 to 8, 12, 16, 20, 30, 40, 50 and 60.

We also calculated the average RMSE at the same forecasting horizons but gave the RMSE a sign depending upon the predicted GDP's trajectory below (positive values) or above (negative values) the observed GDP. The RMSE varied between −0.07 and 2.0. The

predicted GDP was above and below the observed GDP about 50% of the time, but the cumulative RMSE is positive except at time horizon 30 months.

### 4.2. The Recession Forecasts

In Table 3, we compare the root mean square errors for the pre-recession forecasts with a random set of uniform stochastic series. The last column of Table 3 shows that the RMSE for the comparison varies 3-fold. Equation (e), Table 2, was not significant at the 95% level and is not included in Table 3. Among the five recessions, the early 2000 recession is predicted the best, and the 1990 recession is predicted the worst. All equations, except GDP = f(T), show RMSE values less than the test statistics. Visually, the equation GDP = f(T, $T^2$, FF) shows the sharpest peaks, caused by the second-order term in T. For the recession of 2008 (the out-of-sample recession), all equations predicted the recession better than the test statistics.

**Table 3.** Root mean square errors for the recession forecasts as percent of the test values in the rightmost column.

| 20M before Recessions | f(T) | f(T, FF) | f(Tw) | f(Tw, FFw) | f(T, $T^2$, FF) | Aver. | St.dev. | Random RMSE |
|---|---|---|---|---|---|---|---|---|
| Row no. in Table 2 | a | b | c | d | f | | | |
| 1978M6–1981M1 | | | | | | | | |
| 1980 Recession | 9 * | 13 * | 82 | 62 | 31 * | 40 | 32 | 2.25 ± 0.67 |
| 1979M12–1981M7 | | | | | | | | |
| 81–82 Recession | 51 ** | 4 ** | 7 ** | 40 ** | 19 ** | 24 | 21 | 2.36 ± 0.26 |
| 1988M12–1990M5 | | | | | | | | |
| 1990 Recession | 126 | 64 | 16 ** | 23 ** | 36 * | 53 | 45 | 1.43 ± 0.26 |
| 1999M6–2001M3 | | | | | | | | |
| 2000 Recession | 3 | 14 * | 42 * | 2 * | 18 * | 16 | 16 | 1.54 ± 0.38 |
| 2006M5–2007M12 | | | | | | | | |
| 2008 recession | 25 * | 13 ** | 71 | 21 ** | 35 * | 33 | 23 | 4.15 ± 0.76 |
| Average | 43 | 21 | 44 | 30 | 28 | 33 | | |
| Standard dev. | 50 | 24 | 33 | 23 | 9 | | | |

Note: This table shows root mean square error, RMSE, as a percentage of the test RMSE, with 20 months before the first month of the five recession periods 1977 to 2019. Data for five forecasting equations. The right-hand row shows the RMSE when one series is from the GDP and the other series is a uniform stochastic series. * shows that the RMSE is less than the test RMSE − 2 × St.dev (≈95% confidence interval); ** shows that the RMSE is less than the test RMSE − 4 × St dev.

We applied Equation (b), GDP = f(T, FF), to the full series, the test set and the recessions. For the full series, we obtained an RMSE of 0.975, which corresponds to 73% of the test statistics (0.975/ 1.334 = 0.73). For the test set, we obtained an RMSE of 0.356, which corresponds to 30% of the test set statistics (0.356/1.18 = 0.30). The 2008 recession (in the test set) obtained an RMSE of 13% of its test statistics, and all five recessions obtained an average RMSE of 21% of their test statistics.

### 4.3. Robustness

Although the prediction skill can be good over a certain period, economies often show breakpoints that change the conditions for high-skill predictions. The calculations for robustness across GDP breakpoints showed that the function GDP = f(T, FF) obtained the highest score, 1.5, and the function GDP = f(T, $T^2$, FF) obtained the next highest score, 0.96 (range 0–1.5).

### 4.4. Coparison to Alternative Forecasting Methods

To examine the discrepancy between the high skill of the inverted T and the much lower skill when T is not inverted, we compared T to the Michigan consumer sentiment index that is supposed to forecast the short- and long-term (1 year ahead) outlook for the GDP. Sentiment indexes are used extensively both as leading indexes that are published separately, http://www.sca.isr.umich.edu/, accessed on 20 March 2021, and as a component in sets of predictors, e.g., Fornaro [25]. The two curves for GDP = F(T, FF) and MCSI

are counter-cyclic in 76% of our study period, that is, from 1980 to 2009 and from 2012 to 2016. The MCSI is a leading index to GDP 13% of the time during the period 1977 to 2019 (Supplementary Materials 2). The inverse relation between the MCSI and the yield curve seems to relax just after the 2008 recessions.

## 5. Discussion

We used two sets of the independent variables T and FF: first, an unrestricted set, and then a set restricted to the time windows where the two variables were leading variables to GDP. Our main argument for restricting the time window is that information is then available in advance.

We first compare the predictions of the six equations that were calibrated on the learning set to the observed GDP for the full sample period 1977M1 to 2019M5. Then, we compare predictions to observations for the test set 2005M1 to 2019M5. Last, we compare predictions to observations for the 20 months prior to the five recessions. Note that it is only the last 2008 recession that belongs to the test set. We finally discuss how the results relate to the four hypotheses in the introduction.

### 5.1. Comparison Metrics

We compare our predictions by using the regression coefficient R (and the explained variance $R^2$) and by comparing the RMSE of our predictions to stochastic time series. However, the RMSE tends to give a higher (worse) score and the least square (LSQ) methods will give a lower (worse) score the longer the predicted curve is shifted relative to each other. (Two sine functions that are shifted $\frac{1}{4}$ of a common cycle length relative to each other give a regression coefficient of r = 0.) Still, the predicted and the observed time series may be close replicates of each other. Thus, the prediction skill measure depends on the shift between the prediction and observed time series in addition to their cyclic characteristics. Both R and RMSE metrics are used in the literature, e.g., R (and a pseudo-R) was used by Estrella and Hardouvelis [1] and Fornaro [25], and versions of the RMSE were used by Gupta et al. [26] and Plakandaras et al. [27].

### 5.2. Forecasting GDP

The observed and the predicted curves are shown in Figure 2, and the test statistics for the full period, the learning period and the test period are shown in Table 2. The test statistics for the recession periods are shown in Table 3. The regression coefficient, R, was positive and above the adjusted $R^2 = 0.426$ for all forecasting horizons less than 12 quarters using GDP = f(FF, T) as the prediction algorithm. Estrella and Hardouvelis [1] found an optimum adjusted square, $R^2 = 0.44$, for forecasting horizons of six and seven quarters. Thus, the predictions for the period 2005 to 2019 were overall better than those for the period 1955 through 1988 studied by Estrella and Hardouvelis [1].

We found that Equation (f): GDP = f(T, $T^2$, FF), and Equation (d): GDP = f(Tw, FFw), gave the largest adjusted explained variance. However, the lowest RMSE values for the out-of-sample forecasts were obtained with Equation (b): GDP = f(T, FF). This equation, and Equation (f), GDP = f(T, $T^2$, FF), gave the longest time period between the forecasted turning point and the recession turning point for the 2008 recession in the test set (7 months). Bauer and Merten [19] reported that the delay between the term spread turning negative ranged between 6 and 24 months. Our initial calculations gave 17.5 ± 3.1 months for the five last recessions.

Visual inspection of the predicted and the observed GDP series in Figure 2a and e shows that the peak that appeared just after 2010 in the GDP = f(T) graph seems to be due to the T since it is not present in the GDP = (FF) graph. Thus, the graphs in Figure 2 suggest that the optimum forecasting horizon depends on the actual development of the GDP during the test set period and on the forecasting function used, e.g., the false peak caused by T in 2010.

The second-order term in T appears to have four effects on the predicted GDP. The multiple regression (f) that includes $T^2$ explains a larger proportion of the explained variance than the regression (b) without $T^2$. This makes the predicted GDP curves more peaked. This fits well with the peaks around 2000 and 2008, but not with the 1990 peak and the recovery period that led up to that peak. It appears to be responsible for an apparent recovery period in 2009–2010 just after the 2008 recession that did not occur. Finally, the RMSE-T of the out-of-sample predictions was larger than that for all the other alternative equations, except for Equation (e), with only FF as an argument.

*5.3. Recession Periods*

To obtain a fuller picture of the forecasting skill of the equations, we also examined the four recessions that belong to the learning set. The 1981–1982 and 2000 recessions were overall the best predicted (smallest RMSE), and the 1990 recession was predicted the worst, Table 3. The relatively poor prediction during the 1990 recession agrees with the results by Estrella [28] and Ang, Plazzesi [16], who found that during the post-1987 period (the Volcker/Greenspan period, McNown and Seip [29]), the predictive power of the yield spread was diminished probably because of the Volcher strict inflation targeting of the monetary policy.

The RMSE statistics are susceptible to shifts between cyclic series. We used a relatively large time window that led up to the recession (20 months) to evaluate the relation between the observed and the predicted values. This would allow for some uncertainty in the time of the recession prediction. For the latest out-of-sample recession in 2007M12, the timing between the observed peak and the predicted peak was, on average, 5 months, ranging from 1 to 7 months depending on the equation used.

Visually, the two last recessions in 2000 and 2008 formed the sharpest peaks, that is, the recovery from the previous recession to the 2000 and 2008 recessions was relatively short. However, employment growth was relatively slow [30]. The 1990 recession was, in contrast, preceded by a volatile period. The overall best predictor for all recessions is the GDP = f(T, FF) equation. It gave an overall RMSE of 21% of the test statistics for the recessions. GDP = f(T, $T^2$, FF) came in second place. The only equation that gave a worse result than the reference statistic was GDP = f(T) for the 1990 recession, Table 3.

*5.4. The Hypotheses*

We now discuss our four hypotheses.

**Hypothesis 1 (H1).** *The first hypothesis, that T alone would explain most of the variation in the detrended GDP, was not supported; three other alternatives gave a lower RMSE for the test set and for the recession forecasts,* Tables 2 *and* 3*. Meanwhile, the treasury yield curve T itself appears only to have a high predicting skill when it obtains a negative value. Flattening of the T did not reflect a substantial increase in the probability of a near-term recession. A reason may be that the Fed's monetary policy affects both the FF and the T. Lowering the target FF in anticipation of a coming slowdown may increase the slope of the T [20].*

**Hypothesis 2 (H2).** *There is long tradition for adding variables in the predicting algorithm. Our second hypothesis, to augment the T with the FF, was supported, but only the second-order term in T enhanced the forecasting skill, and only under certain circumstances. The use of "big data techniques", that is, to expose large datasets to data selection techniques, such as PCA or machine learning techniques, e.g., Stock and Watson [31] and Medeiros et al. [22], respectively, can assist in evaluating variables. The machine learning technique allows modeling of nonlinearities, and the technique used by Medeiros, et al. [22] computes principal components that include only the variables that show a high prediction power (the author's target variable is inflation). A second option is to use economic insights to evaluate candidate variables. Variables that have proven to robustly contribute to high prediction skill for inflation are prices, housing prices and employment [22]. The conference board of the USA uses a composite leading index (CLI) with ten components. Among*

them is the interest rate spread, T, that has the federal funds rate included, and an average consumer expectation for business conditions, but not the federal funds rate as an independent component [32]. Bauer and Merten [19] showed, using a probit model, that its predictive power is largely unaffected by including additional variables, e.g., estimates of the natural level of interest or household net worth-to-income.

Our prediction function, GDP = f(T, FF), and the MCSI had quite different characteristics with respect to pro- and counter-cyclicity, and the MCSI was a leading index only during 13% of the time in the period it was applied to. We do not have any explanation for this. In contrast, the German leading indexes ife (lower case letters) and ZEW that are based on managers' and economists' sentiment for changes in IP were leading IP 77–78% of the time (Seip, Yilmaz et al. 2019). A "horse race" between the MCSI and the forecasting function GDP = f(T, FF) showed that the MCSI was a leading index to GDP 14% of the test period, whereas f(T, FF) was leading 26% of the time. However, for the first year of the test period, 2005, f(T, FF) was leading 50% of the time and MCSI 6% of the time.

We provided rationales for introducing interaction and second-order terms for T and FF in the introduction. We found that the second-order term, $T^2$, could give a better prediction, but only under certain circumstances (the 1990 recession). The generality of those circumstances is not known, but it occurred during the period that is named "The Great Moderation".

**Hypothesis 3 (H3).** *Lead–lag. Lead–lag relations. A variable must be leading the target variable and therefore must be shifted relative to the target to allow measurements of its skill in predicting the target. Medeiros, Vasconcelos [22] examined four lags (months) for all their candidate variables. However, a process such as hiring or shedding employees may take longer but may still have predictive power. Our third hypothesis, that restricting the regressions to time windows where T and -FF were leading variables to GDP would improve predictions, was not supported. For the test set, the RMSE for the best prediction f(T, FF) increased substantially when restrictions were applied, Table 2, Equations (b) and (d). Predictions for the recessions increased the RMSE from 21 to 30 percentage points. The reason may be that the portions of the time series where T and -FF were not leading GDP do not affect the forecasting skill much.*

**Hypothesis 4 (H4).** *Recessions and breakpoints in GDP. A crucial question is whether the economy has evolved so that variables have changed their predictive power. Structural breakpoints in the US economy were identified by Perron and Wada [33] and McNown and Seip [29]. Dates during the studied period are 1975M6, 1979M2, 1983Q4 (the start of the "Great Moderation" period 1983Q4 to 1997M2), 1991Q4, 1999Q3 and 2007M4. Our fourth hypothesis, that recession periods would be predicted better than the full time series, was supported (with 21% vs. 73% of the respective RMSE). This result is also supported by Hassani et al. [34] studying the 2008 recession during the period 2000 to 2010 with the singular spectrum analysis technique. They found that the average RMSE estimates relative to their benchmark model were for pre-recession (2.11), the recession period (7.11) and the post-recession period (5.60) (their leading series 1–4). Rudebusch [17] suggested that the predictive power of the inverted T endures, whereas Schrimpf and Wang [14] showed that the predictive power weakened substantially after 1984. Johansson and Meldrum [20] suggested that the flattening of the T in the past years is due to a slower expected GDP growth. Seip and McNown [35] calculated a robustness score based on timing and accuracy and found that the CLI (robustness = 4.0) was, on average, more robust over time than the average working hours (AWH) (robustness = 3.1); however, AWH beat the CLI (robustness = 6.4 to 1.4) during the period 1988:2 to 2006. A contrasting result was found by Glosser and Golden [36], who found that AWH has been less associated with the business cycles after the breakpoint in 1979. Generally, economic breakpoints may alter the conditions for a prediction algorithm to show high prediction skill. For two German sentiment indexes, prediction skill was weakened during periods with abnormal economic states [15].*

Finally, we discuss the lead time. The actual FF values become available when the Fed determines the short-term rates. However, discussions that the Fed may have before FF is actually determined may be available in advance of the actual values [37]. We found a phase shift for T vs. GDP of 15–20 months. This is a little longer than the phase shift of 12 months used by, for example, Ang et al. [18] in their Fig 2, and the lead times cited by Rudebusch and Williams [38]. We also tried to lag the GDP relative to the FF and the T, but in contrast to Ponka [39] and Wang, Nie [40], lagged variables did not improve the predictions.

## 6. Conclusions and Policy Implications

Our study shows that both the treasury yield curve (T) and the federal funds rate (FF) are important indicators for future economy development. To make a prediction for a possible coming recession, one of the forecasting functions, GDP = f(T, FF) or GDP = f(T, $T^2$, FF), should be applied to a learning set. If the forecast predicts a recession, then the real recession may come at the forecasted recession time plus 5 to 7 months. Our result contrasts with the high prediction skill of the T when it obtains a negative value. However, if the T is approaching a negative value, but it is not yet known if it will actually reach it, then the use of forecasting functions such as GDP = f(T, FF) or GDP = f(T, $T^2$, FF) may be beneficial. Furthermore, if there is a choice between the forecasts of a consumer sentiment index or the GDP = f(T, FF) forecast for the general development of the economy, the latter should be preferred. A second issue, as pointed out by Akerlof and Shiller [41] and Andolfatto and Spewak [42], is that the economic state at the time of a suspected recession may be susceptible to shocks, and the actual recession may, or may not, be triggered by the shock. A question is therefore if investors are better at detecting non-rational behavior than other decision-makers, or if they are triggered by a negative T. Second, recessions appear to be easiest to predict if there has been a long and uninterrupted increase in GDP, such as the increase before the 2001 recession.

Our study rests on several assumptions that can be questioned, some of which have been addressed in the Discussion section. Three notable issues are the use of linear detrending, the effectiveness of using R or RMSE as measures of forecasting skills and the role of anomalies in the economy for the forecasting skill obtained. This study shows that the forecasting skill differed among recession periods. An area of future research is therefore to identify, if possible, the economic characteristics of the time windows leading up to each recession and the reasons for differences in the prediction skill among recessions.

**Supplementary Materials:** The following are available online at https://www.mdpi.com/article/10.3390/forecast3020025/s1, Figure S1: Lead–lag (LL) relations for the test set, Figure S2: LL relations between the yield curve and the Michigan sentiment index, US economy.

**Author Contributions:** Conceptualization, K.L.S. and D.Z.; methodology, K.L.S.; software, K.L.S.; validation, K.L.S. and D.Z.; formal analysis, K.L.S.; investigation, K.L.S. and D.Z.; resources, K.L.S. and D.Z.; data curation, K.L.S. and D.Z.; writing—original draft preparation, K.L.S.; writing—review and editing, K.L.S. and D.Z.; visualization, K.L.S.; supervision, D.Z.; project administration, K.L.S.; funding acquisition, K.L.S. and D.Z. All authors have read and agreed to the published version of the manuscript.

**Funding:** This research was funded by Oslo Metropolitan University.

**Informed Consent Statement:** Not applicable for studies not involving humans.

**Data Availability Statement:** All data are available from the first author.

**Conflicts of Interest:** The authors declare no conflict of interest.

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
