# Peer review of "The Yield Curve as a Leading Indicator: Accuracy and Timing of a Parsimonious Forecasting Model"

_forecasting, doi:10.3390/forecast3020025_

Round 1

Reviewer 1 Report

The paper needs improvements in all aspects. Please consider shortening the paper, including latest references in the literature, and specifying appropriately, the model that are estimated, and clearly explaining the estimation procedures, and highlighting contribution of the research with respect to the recent studies in this area. 

Author Response

See enclosure

Reviewer 2 Report

The paper tweaks standard regressions models to forecast US GDP growth and NBER-dated recessions and show that expanding the model to include the federal funds rate and square predictors, including the Fed funds rate itself. Empirically, the paper achieves some success, even though the practice of qualifying which variable improves the prediction of which recession is a bit odd, not if the key result of the paper can only be stated in these terms. Some of the results claimed as added value are a bit at the border of inducing a smirk, see for instance "we show visually how time series predicted with [the term structure]  alone would differ from predictions with FF alone." -- is not natural to do so? Is that a big deal? 

In fact, my most serious concern is that the manuscript may be not doing enough -- apart from a plain vanilla, MSc-level contribution -- to clear the bar at Forecasting, even though this is a new journal. For instance, you are right that your "(...) forecasting model that combines parsimony, predictive 
accuracy, and ease of estimation", but it is almost obvious: it is a regression! I suggest two reactions. First, given its results the paper could be shortened 3-4 pages (before any additional empirical results) keeping all empirical results just by saving on the literature review on a few rather self-evident claims that do not really help the paper to showcase its contribution and give it instead some MSc. flavor that needs to be purged at all costs. Second, additional analyses must be performed.

The paper lacks a bit in motivation: if one is already forecasting with the term spread, which obviously includes at least one measure of the short-term rate, adding the Fed funds rate "because the Federal Reserve’s policy is known to influence both the treasury yield curve and the future economic growth." sounds a bit trivial, almost childish -- what is the point really here? Why adding (near-) I(1) variables on top of a stationary predictor (the term spread) may improve things?

Why to perform a comparison of yield curve-based predictors in particular with the Michigan consumer sentiment index (MCSI)? Why this one in particular? Here it would make sense to engage in a comparisong with factor (FAVAR) models instead, I feel, see e.g., Fornaro (2016, J. of Forecasting) or Gupta et al. (2017, J. of Forecasting).

At p. 13, H-2 the paper mentions nonlinearities but then I cannot really follow what happens and what the empirical work may be to assess such a hypothesis. My question is then simple: is it possible that simple regime switching model (threshold or Markov, all currently implemented in Eviews for instance) JUST BASED ON THE TERM SPREAD may outperform your operation of adding just some rates or their squares? Although related to asset return forecasting, Guidolin and Ono (2006, JEB) provide a few example of how that simple strategy may often outperform.

If the goal is to predict recessions, how come there is no (dynamic, conditional) probit-style methods applied in this manuscript? Footnote 4 makes no sense to me: either this is a forecasting paper or not and there are at least 20 published papers with probit-based forecasting of recessions. Full stop. 

The paper needs considerable revision as far as grammar and style are concerned. A few examples follow,  but the problem is general and careful editing and polishing will be crucial to a successful revision.

(Abstract) The reference to "(...) the movement of future economy." needs re-writing and is currently ambiguous. Same applies to "(...) by being a leading index to the observed GDP for longer  periods (6% vs. 50% of the time for the first test year).", also in part that refers to numbers (what are they?)
(Keywords) I am stunned by the keywords: "keyword 1; keyword 2; keyword 3 (List three to ten pertinent keywords specific to the article yet reasonably common within the subject discipline.)", that is odd...
(p. 1) To provide evidence internationally literally means that the evidence is sent to the international community, please re-phrase.
(p. 2) I cannot understand the expression TO BE SUBJECTED to non linearities -- subjected by whom? Same for CONTAINING GREAT INFORMATION, this is not standard academic style in English.
(p. 2) Forecasting SKILL? Power? I cannot follow.
(p. 2) I cannot follow the use of TO REPLY in "For instance, policy-makers may reply on the forecasts to justify their policies for economic expansions and contractions."
(p. 3) I cannot understand how the possessive case is used in "Rudebusch and Williams (2009)’ findings are robust in all dimensions (...)".
(p. 4) What are the MATERIALS. Some words are used out of context as if this may be the result of some automatic translator application? Please revise, that needs attention.
(p. 4) Why is estimation called CALIBRATION at some point? In an applied econometrics outlet? Later, is to DETERMINE the coefficients the same as estimating them? That was very awkward.
(p. 5) I cannot understand the meaning of "To shift the regressors backward, we apply a lead-lag method (Seip, Yilmaz, and 200 Schroder 2019, Seip and McNown 2007) to the time series to identify time windows where 201 the regressors lead the target variable GDP."
(p. 6) Is it common to work with "normalized the series to unit standard deviation" (aka standardized, I guess) in this literature?

Author Response

See attatchment

Reviewer 3 Report

The authors have identified the research gap correctly. The hypotheses posed in the paper are reasonable. There are however serious flaws in the description of the methodology. It is not clear how the yield curve was modeled and whether multiple regression between yield curve and GDP is enough to model possibly complex relationships between nonstationary data.

Author Response

See enclosure

Round 2

Reviewer 3 Report

I feel satisfied with the improvements. The revised version of the paper is well organized. Authors correctly point that empirical time series are not generated from a normal distribution, and that time series representing yield curve shape include some sort of seasonality.

Minor language and style improvements might be needed.